# How to Detect Online Hate towards Migrants and Refugees? Developing and Evaluating a Classifier of Racist and Xenophobic Hate Speech Using Shallow and Deep Learning

Carlos Arcila-Calderón [1,*] , Javier J. Amores [1] , Patricia Sánchez-Holgado [1] , Lazaros Vrysis [2] , Nikolaos Vryzas [2] and Martín Oller Alonso [3]

1. Facultad de Ciencias Sociales, Campus Unamuno, University of Salamanca, 37007 Salamanca, Spain
2. Multidisciplinary Media & Mediated Communication Research Group (M3C), Aristotle University of Thessaloniki, 54124 Thessaloniki, Greece
3. Department of Social and Political Sciences, Università degli Studi di Milano, 20122 Milano, Italy
* Correspondence: carcila@usal.es

**Abstract:** Hate speech spreading online is a matter of growing concern since social media allows for its rapid, uncontrolled, and massive dissemination. For this reason, several researchers are already working on the development of prototypes that allow for the detection of cyberhate automatically and on a large scale. However, most of them are developed to detect hate only in English, and very few focus specifically on racism and xenophobia, the category of discrimination in which the most hate crimes are recorded each year. In addition, ad hoc datasets manually generated by several trained coders are rarely used in the development of these prototypes since almost all researchers use already available datasets. The objective of this research is to overcome the limitations of those previous works by developing and evaluating classification models capable of detecting racist and/or xenophobic hate speech being spread online, first in Spanish, and later in Greek and Italian. In the development of these prototypes, three differentiated machine learning strategies are tested. First, various traditional shallow learning algorithms are used. Second, deep learning is used, specifically, an ad hoc developed RNN model. Finally, a BERT-based model is developed in which transformers and neural networks are used. The results confirm that deep learning strategies perform better in detecting anti-immigration hate speech online. It is for this reason that the deep architectures were the ones finally improved and tested for hate speech detection in Greek and Italian and in multisource. The results of this study represent an advance in the scientific literature in this field of research, since up to now, no online anti-immigration hate detectors had been tested in these languages and using this type of deep architecture.

**Keywords:** hate speech; racism; xenophobia; migration; social media; deep learning

## 1. Introduction

Violent speech is not an exclusive communicational dysfunction of our contemporary societies, but it is today, when it seems more worrying than ever due to its massive diffusion on digital platforms. The internet and information and communication technologies have today allowed online hate speech to increase unabated. In this new context, social media has become the forum in which this type of message spreads more quickly and uncontrollably, as evidenced by the latest reports published by the Anti-Defamation League [1,2]. This growth in online hate speech also coincides with an unstoppable increase in registered hate crimes in Europe [3], which could evidence the correlation between both phenomena pointed out by Müller and Schwarz [4]. Moreover, if this connection is so, since most of the hate crimes committed in Europe are due to racist and/or xenophobic reasons (according to the data collected by the OSCE's hate crime reporting), we could affirm that most of the increasing hate speech that is spread online is based on this type of discrimination

and is aimed mainly towards migrants, refugees, and asylum seekers who come to or are within European borders. In this same line, recent works developed by the authors have evidenced a negative trend in the representation of migrants and refugees that is spread by the main media of Mediterranean countries [5] and in Western Europe [6], which could be also related to the increase in racist and xenophobic hate. Other studies also indicate the phenomenon of deresponsibilisation [7] of hate speech spreading online, especially by younger audiences who feel that their public hate language should not be taken seriously.

With these premises, some researchers have understood that is urgent to explore new methods for detecting and preventing online hate at the global level, but also in regional contexts, where online and offline hate has not stopped increasing either. For this reason, in recent years, diverse public and private institutions have been making great efforts to try to detect and counter hate speech online, although mostly in a general way and not dealing specifically with racist and xenophobic hate. In addition, the large amount of information offered on digital platforms today makes it more difficult than ever to monitor, detect, and combat these hateful contents. This, in turn, means that victims of online hate might be increasing, something that, in the Spanish case, the latest Raxen reports [8,9] show, even though most incidents might not be recorded. In this situation, it is important to try to develop new methodological strategies that allow us to monitor these violent speeches that spread on social platforms, paying special attention to racist and xenophobic hate. Taking this into account, it is surprising that, although there are already several researchers who are addressing this problem in English, there are still so few researchers who are doing it in other speaking contexts, focusing specifically on anti-immigration hate online, the category that internationally worries the most [10,11].

With these premises, the aim of this work is to generate and test a detector of racist and xenophobic online hate speech. The novelties of this work compared to the previous ones are several. In the first place, for the development of the prototypes, we will generate ad hoc datasets with messages extracted from social media, manually annotated by several trained coders. Second, in addition to shallow learning algorithms, deep architectures will also be used in the development of the prototypes. On the one hand, an ad hoc developed RNN model is developed, and on the other hand, a BERT-based model is developed, in which transformers and neural networks will be used. Finally, although the prototypes will be trained and tested first in the Spanish language, the ones with the best performance will later be improved and retrained following the same strategy so that they are capable of detecting anti-immigration hate speech also in Greek and Italian (Mediterranean countries that are the main gateway for most immigrants arriving in Europe) and in multiple sources. These detectors will identify increases in this type of cyberhate and develop tailored programs to combat and counter it, but also will acquire empirical knowledge about these violent speeches, about the groups to which they are addressed, about the sources or profiles propagating hate, and lastly, about how these types of messages could be triggering hate crimes in the physical environment. In short, this prototype will allow us to understand the spread of online hate aimed towards displaced people and to devise strategies to counteract and prevent its possible effects, including physical hate crimes.

## 2. Defining Online Hate Speech

Hate speech is not a new concern; in 1997 Calvert already pointed to this type of discourse as a problem to analyse, understand, and combat with communicational approaches, necessarily involving all elements of the communication transmission models. However, hate speech has become a bigger concern today due to the rapid growth of digital media, especially social media, in which former readers and audiences have become prosumers [12], with more and more followers to whom they can launch their messages and content. In addition, nowadays, it seems that the more sensationalist this content is, the more followers it gets. Without a doubt, in this new dimension of immediacy and freedom, it is much easier for hateful messages to spread quickly and without any kind of control. For this reason, hate speech has not stopped increasing in recent years, and

that is why it has become something so complex and difficult to detect and combat, as well as necessary and urgent. This is what the present work tries to solve through a novel computational strategy that seeks to automatically detect latent hate in the messages spread through Twitter, primarily in Spanish and, secondarily, in Greek and Italian. However, before tackling any online hate speech detection strategy, it first needs to be defined.

Thus, from a theoretical approach, hate speech is understood as the promotion of messages that imply rejection, contempt, humiliation, harassment, discrediting, and stigmatization of people or social groups based on attributes, such as nationality or colour of skin. Thus, for speech to be considered hateful, one of the main conditions is that the discriminatory message is directed towards one of the vulnerable groups typified in the European framework, or towards an individual who can be identified as part of one of those collectives, whose rejection is motivated by their apparent belonging to the group. In this sense, the Council of Europe [13] adds that for speech to be understood as a hate crime, it must propagate, incite, promote, or justify racism, xenophobia, anti-Semitism, and other forms of intolerance. The European Commission against Racism and Intolerance, in its General Recommendation No. 15 [14], also specifies that hatred can be motivated by reasons of race, colour, ancestry, national, or ethnic origin among many other characteristics or personal conditions. As can be seen, the official definitions of hate speech pay special attention to racist and/or xenophobic discrimination as the main cause of all types of rejection and hate. For its part, the Ministry of the Interior of Spain, in its latest evaluation report on hate crimes in Spain [15], collected a total of 11 categories of discrimination into which crimes committed against vulnerable audiences can be classified, where racism and/or xenophobia are the first, in which more crimes are registered every year.

In the academic sphere, some authors such as Miró Llinares [16] have also studied hate speech offering, in addition to a broad definition, a taxonomy with different levels of hate online. Thus, according to this author, it is possible and necessary to differentiate between the type the hate speech that could constitute a crime, from the speech that, even expressing rejection and intolerance of certain vulnerable groups, can be framed within the margins of freedom of expression. These types of messages would include slight insults, criticism, and offenses to individual or collective sensitivity, which in some cases, could be an attack on people's dignity, but not a hate crime. Regarding illegal hate speech, these types of messages would include all those that are spread in a public and massive context and that more directly and explicitly incite violence, intimidation, hostility, or discrimination against a vulnerable group or an individual belonging to a vulnerable group—in the case of racist and/or xenophobic hate, they would be migrants, refugees, asylum seekers, and all kinds of stigmatized races, ethnicities, and nationalities. With these premises, the present work covers all the typified levels of hate, trying to extend hate speech detection as much as possible, considering that the most explicit hate (which could be considered a crime) is considerably reduced in European contexts. However, it is expected that in the training process and in generating the machine learning models, the final prototype will be refined, and detection will finally be limited to the most explicit levels of hate.

## 3. Detecting Racist and Xenophobic Hate Speech Online

In recent years, many authors have studied hate speech online from very different perspectives. Chetty and Alathur [17] analysed it from the jurisprudential basis, concluding that appropriate political measures as well as the actions of social platforms are essential to effectively counteract hate speech. Other authors, such as ElSherief et al. [18], analysed it using a data-based linguistic and psycholinguistic perspective, offering a framework of understanding from which to identify the hate that is spread on social media. With a more automated and massive detection approach, Mondal, Silva, and Benevenuto [19] proposed a system for measuring and monitoring hate speech propagated on the social networks Twitter and Whisper based on specific keywords and expressions, focusing on the recognition of the main targets to which hate is directed massively. For their part, Malmasi and Zampieri [20] as well as Salminen et al. [21], are some of the few authors who proposed

methods to automatically detect hate spread on social media based on NLP and supervised classification techniques.

However, all these works have something in common: they all deal with hate speech from a generic and international point of view; that is, trying to identify hate speech spread just in English, motivated by all kinds of discriminatory reasons, aimed towards all types of vulnerable audiences, and at any time and context, is an approach that is too ambitious and could pose a problem of internal validity, especially in large-scale strategies. Even the prototype recently developed by Salminen et al. [21], one of the most innovative and advanced prototypes using deep learning and including detection in various online sources, is based on this same type of approach. Trying to detect online hate in a general way can be reductionist by obviating the complexity of how hate speech is spread, trying to cover them all in a single classifier trained with general examples. This could be a limitation because the resulting models may not be as effective, reliable, and, paradoxically, generalizable as those that are trained with real examples of a specific context, a specific type of hate, and a specific discriminatory category, separating and differentiating concepts, characteristics, and linguistic nuances.

In this sense, it should be noted that on the international scene there are already some examples of strategies and tools for detecting cyberhate that take into account the different levels of hate speech, as well as some of the different categories of prejudice that can motivate it or the different vulnerable groups who may be victims. We can highlight works such as the one developed by Davidson et al. [22], which differentiates between messages that express explicit hate and messages that are just offensive, or the one developed by Badjatiya et al. [23], which aims to specifically identify messages with racist or sexist content and also uses deep modelling. However, most of the cited studies that offer automatic hate speech detection methods based on machine learning have another limitation in common: they do not use ad hoc generated training corpus. Most of the prototypes developed so far base their detection on previously developed lexicon dictionaries, or, in the case of using a corpus of examples to train the classification algorithms, they use already available datasets developed in previous works, such as the prototype developed by Salminen et al. [21]. This approach also influences the internal validity of the prototype and its final reliability. In the Spanish context, one of the few studies that attempted to address the detection of online hate speech in Spanish is the one developed by Pereira Kohatsu et al. [24]. This prototype presents the same limitations as most of those developed internationally since it also addresses hate speech in a generic way without distinguishing audiences or types. In addition, although Pereira Kohatsu did develop an ad hoc training corpus to generate predictive models, this corpus was generated by a single coder, which also poses an internal validity problem due to its potential subjectivity. Similarly, there are recent projects that have created corpora of hateful speech in the Greek [25,26] and Italian languages [27] for the training of hate speech detection models.

From a more technical standpoint, recurrent neural networks (RNN) have become a popular choice for hate-speech detection and classification in short micro-blogging texts [28,29]. Duwairi, Hayajneh, and Quwaider [30] investigated the ability of convolutional neural networks (CNN), CNN-LSTM (long short-term memory), and bidirectional LSTM-CNN models to detect hateful content from social media in the Arabic language, with the last two architectures combining CNN and RNN achieving the best scores. In the work of Al-Hassan and Al-Dossari [31], a support-vector machine (SVM) classifier is compared against LTSM, CNN with LTSM, a gated recurrent unit (GRU), and CNN with GRU models. All deep learning models outperform the baseline, with the combined architecture achieving better performance in this case as well. Al-Makhadmeh and Tolba [32] propose an ensemble of deep classifiers combined with a natural language processing (NLP)-based semantic feature extraction layer. Prasad and Mishra [33] explore the feasibility of bidirectional encoder representations from transformer (BERT)-based models for multilingual hate and abusive speech detection, one of the most advanced techniques in this line which has also been tested in this work. In [34], transfer learning is investigated, introducing

the BERT-based transformer, AraBERT, that shows an improved performance in Algerian dialectal Arabic. In [35], multitask learning (MTL) is proposed for the adaptation of a pretrained hate speech detection model in the Arabic language in cross-corpora tasks. Hate speech detection models are adapted to the target domain, and their performance deteriorates significantly when applied in different domains [36]. This is also shown in the work of Bashar, Nayak, Luong, and Balasubramaniam [37], who trained models for hate speech detection in the context of the COVID-19 pandemic. The importance of the dataset size and quality is highlighted also by Kovács, Alonso, and Saini [38] in both classic machine learning and deep learning approaches. Text preprocessing can also significantly improve performance [39].

Considering these premises, the general objective of the present work is to develop and validate a more advanced computational strategy that allows for the detection of hate speech online based on racism or xenophobia following the lines of research that the authors have already been developing specifically in the Spanish contexts, which could be considered as pilot studies on which this project is based [10,11,40,41]. In most of these studies, the authors treat racism and xenophobia as a single category in the same way that the Ministry of the Interior of Spain does when it records hate crimes. This is based on the fact that both types of discrimination are parallel and present difficulties to differentiate. According to authors such as Díez Nicolás [42] or Cortina [43], on many occasions, even with help of measurement tools, it is too difficult to distinguish between one and the other type of prejudice as the main reason for rejection and hate, since in most cases, the categories are concatenated, intertwined, and one is intrinsically linked to the other. For this reason, they are usually studied together.

In a more particular way, the present work aims to solve and overcome the limitations of previously developed prototypes based on a series of differentiating elements. On the one hand, we exclusively focus on detecting hate speech motivated by racism and xenophobia, which allows for the elaboration of more specific, complete, and precise corpora to generate more reliable predictive models. In the same way, we generate our own datasets of real tweets, at first, only in Spanish, but later, also in Greek and Italian, to train the predictive models. In this sense, since the creation of these corpora requires the manual annotation of previously downloaded and filtered messages from the Twitter APIs, we pose the following research question: What frequency and percentage of hate tweets due to racism/xenophobia are detected through manual annotation in a sample of previously filtered tweets about migration? (RQ1).

On the other hand, another innovative element that this work presents is the use of deep learning in the generation of predictive models that will allow for the classification of hate in Twitter messages automatically and on a large scale. Specifically, recurrent neural networks will be used (and an ad hoc model and a BERT-based model), an algorithm that, a priori, should present significant advantages over traditional classification algorithms, offering better a performance, especially when applied to text classifications, as is the case. However, there is not enough empirical knowledge to affirm that deep modelling will offer a higher reliability than shallow algorithms. For this reason, we also pose the following questions: Which machine learning algorithm presents the best performance when generating a predictive model capable of detecting hate speech spread on Twitter in Spanish, based on racist/xenophobic reasons (RQ2)? Does deep modelling perform better than shallow modelling for generating a prototype capable of detecting racist/xenophobic hate speech on Twitter in Spanish (RQ2A)?

In addition, this work includes an external validation phase as another innovative element in which the first developed classifier is tested with a new sample of tweets. This stage will check, beyond the internal evaluation of the prototype, how reliable the model with the best evaluation metrics is when it comes to detecting new messages in Spanish about migrants and refugees posted on Twitter. Moreover, regarding this stage of validation, we pose the following research question: Will the best performing algorithm reliably detect hate speech in a new sample of tweets about migration in Spanish (RQ3)?

Finally, the research also includes the evaluation of the detector in other languages and other sources as well, not only Twitter. Specifically, the models with the highest score validated in Spanish will be trained in additional languages using the same machine learning architecture, that is, messages in Greek and Italian promoting hate speech about migrants and refugees found online. Therefore, one more research question can be posed: Can the best performing machine learning models in Spanish be retrained and applied to other languages as well, keeping the same level of performance (RQ4)?

## 4. Method

As indicated, the detector of racist and xenophobic online hate speech has been developed following a large-scale detection strategy based on the intensive computation of data under the Supercomputing Centre of Castilla y León, Scayle using NLP and machine learning. For this, the methodological work was developed over 4 stages: the initial exploration and theoretical approach, the generation of the datasets, the generation of the predictive models, the external validation of the prototype, and the adaptation of this prototype to Greek and Italian and its evaluation on other sources.

### 4.1. Theoretical Phase

In this phase, we carried out an in-depth qualitative exploration of hate speech that spreads on social media such as Twitter and, specifically, that which is motivated by racist and/or xenophobic reasons. A literature review related to this field of study was also carried out, which served as a theoretical approach. In addition, we identified profiles and hashtags on Twitter through which a greater number of messages containing racist and xenophobic hate are published. Exploring these potential sources of hate on Twitter helped us to better understand and narrow down the different ways in which racist and xenophobic hate is expressed, as well as the different contexts in which it spreads, the most common victims, and the most commonly used terms and expressions. This, in turn, helped us to subsequently generate the linguistic filters that would allow us to download the first sample of potential racist hate tweets for manual classification in order to generate the ad hoc dataset.

### 4.2. Dataset Generation Phase

After the exploratory phase, the dataset had to be created from real and validated examples of short messages containing the type of hate to detect. The objective was that the developed dataset could be used as a corpus to train the hate speech detectors. The generation of the dataset was carried out in a series of sub-phases that are explained below.

#### 4.2.1. Definition and Typology of Hate Speech to Detect

Firstly, criteria were established to define the type of speech to be detected to generate a customized dataset. In accordance with the possibilities that had been identified in the previous qualitative exploration and taking into account both the definitions provided by the different authors and institutions and the European legal framework itself, the definition of hate speech was broadened, encompassing the different meanings and types offered by academia, public institutions, and the Spanish penal code, as well as the three levels of online hate provided by Miró Llinares [16]. Thus, all types of hateful speeches were included for the generation of the dataset, from the most explicit and violent to the most subtle, since in the previous phase, a small minority of directly racist/xenophobic hate had been detected, even in the profiles most polarized. Since the intention was to be able to detect as many messages as possible with racist and/or xenophobic content, it was necessary to cover more types of hate messages, including the most implicit ones. In addition, in the validation process of the manual classification, which would be carried out following the basis of content analysis, and in the subsequent training of the models, it was expected that the results would be refined, leaving only the clearer examples and filtering and rejecting the most doubtful or ambiguous for not having intercoder agreement. For

this reason, it was also interesting to cover the widest possible range of types of hate. On the other hand, what would be considered hate speech due to racist and/or xenophobic discrimination was also defined, compiling all the derogatory terms, expressions, and targets collected in the exploratory phase.

4.2.2. Elaboration of Dictionary Filters and Downloading the First Sample of Tweets

Subsequently, a dictionary of terms and combinations of words were created to serve as a filter for an initial download of potential tweets with racist and/or xenophobic hate. To do this, we started from the qualitative exploration of Twitter accounts, profiles, and hashtags through which a greater number of racist and xenophobic messages are spread in Spain. Thus, Tweets were located using keywords identified in the exploratory phase in which potential victims of this kind of hate were mentioned. They are mainly forced migrants, refugees, and asylum seekers, but also regular immigrants and all types of non-western ethnicities and foreign cultures, racialized people, sub-Saharan Africans, gypsies, Latinos, Asians, Muslims, etc.

Secondly, based on these first examples of messages with hate speech extracted from potential hate perpetrator accounts on Twitter, we made the final selection of the search words [44] to create the final filter dictionary that would be used for the download. Specifically, a list of words, roots, or word combinations that could be representative or indicative of racist and/or xenophobic hate was drawn up. This filter dictionary was developed ad hoc with the aim of accessing tweets most likely to contain the type of hate sought, thus optimizing the tagging, streamlining, and optimizing of the dataset creation process. In this way, the filtered tweets were downloaded, which would later be classified manually to generate the training corpus.

We finally carried out the download using the dictionary generated between October and December 2019. Although we downloaded a larger number of tweets, we finally collected a sample of 24,000 messages for later manual classification.

4.2.3. Manual Pair Classification and Clean-Up of the Final Dataset

After downloading, we proceeded to manually classify the potential anti-immigration hate tweets, for which the Doccano platform was used. All tweets were classified by two binary trained judges as hate and non-hate messages. Simultaneously, the tweets that were not interesting to include in the dataset were discarded, such as those from other contexts, for example. The dataset was generated only with the messages in which there was agreement between both coders, discarding those without agreement. With this step, we intended to ensure the reliability and quality of the resulting dataset, thus overcoming the limitations of some previously developed prototypes, e.g., [24]. After compiling and cleaning the final dataset, it was made up of a total of 3751 racist/xenophobic hate tweets (15.6%) and 7892 non-racist/xenophobic hate tweets (32.9%).

*4.3. Generation of Predictive Models Phase*

In a final phase, we used the dataset developed to generate the classifiers that would later allow us to identify the anti-immigration hate messages spread on Twitter in Spanish. Specifically, we generated a total of nine predictive models, six of them using traditional algorithms, another model from the votes of those shallow models, and two final models using deep learning, specifically recurrent neural networks and transformers.

4.3.1. Shallow Modelling

For the development of the shallow models, the scikit-learn libraries and the Natural Language Toolkit (NLTK) were used. Specifically, 6 models were generated with the following conventional algorithms: original Naïve Bayes, Naïve Bayes for multinomial models, Naïve Bayes for Bernoulli's multivariate models, logistic regression, linear classifiers with stochastic gradient descent training, and a support vector classifier. In all of them, we used the default parameter settings from the scikit-learn library [45] and bag-of-words as the

text representation. In addition, the dataset was randomly divided into two subsets, one with 70% of the messages for training, and another with 30% for testing the models. After training the shallow models, we finally generated a final summary classifier which based its prediction on the votes of the previous 6 classifiers. In this case, a confidence threshold of 80% was included so the summary detector would choose the category predicted by at least 5 of the 6 shallow classifiers.

### 4.3.2. Deep Modelling

Finally, a deep learning architecture was used to generate the final prototype. Specifically, an ad hoc recurrent neural network was generated using embeddings as the text representation. To do this, we used the Keras library with TensorFlow as the backend to create a sequential model with four layers. The input layer was used to create the embeddings, which were trained using the 10,000 most common words of the created vocabulary plus 1000 out-of-vocabulary buckets, as suggested by Géron [46]. Thus, the embedding matrix included one row for each of these 11,000 words and one column for each of the 6 embedding dimensions. The second and the third were hidden layers that consisted of GRUs (a simplified version of the conventional LSTM cells) with 128 neurons each [47]. Finally, the output layer was a dense layer with one neuron and used the sigmoid activation to estimate the probability that a particular message contained racist/xenophobic hate. We used standard loss with binary crossentropy and an Adam optimizer to compile this model. Subsequently, we implemented the training corpus using 10 epochs, and we used the test set for validation (30 steps).

In addition, we developed a final deep learning classifier using bidirectional encoder representations from transformers (BERT) [48], a pre-trained large language model that uses 177,854,978 parameters and that was fine-tuned with our annotated hateful and non-hateful messages. This model was generated using an Adam optimizer with learning rate = $3 \times 10^{-5}$ and epsilon = $1 \times 10^{-8}$, sparse categorical cross entropy for loss, and 3 epochs. The number of parameters of both models can be consulted in Table 1.

**Table 1.** Deep learning algorithms' complexity in terms of models' number of parameters.

| RNN | BERT |
| --- | --- |
| 216,798 | 177,854,978 |

### 4.4. External Validation Phase

Finally, once the predictive models had been generated and evaluated, the classifier with the best performance was validated with new samples. The objective of this stage was to test how accurate and reliable the classifier with the best evaluation metrics is when putting it into practice with new data. This new dataset contained 10,285 tweets retrieved in November and December 2020. The messages were manually classified by two new human coders. In this case, the sample was carefully reviewed before the manual annotation, eliminating before starting to tag all the tweets that we wanted to discard because they came from other contexts or belonged to other categories of prejudice, for example. Thus, after the classification process, only the tweets that did not have an agreement were rejected, with which it was possible to considerably increase the valid tweets with inter-judge agreement. At the end of the classification process, the intercoder reliability was checked again, seeking full agreement. Thus, this manual classification resulted in 83% of the tweets annotated with agreement (n = 8588), of which 2781 were messages of racist and/or xenophobic hate (27.04% of the total, 32.38% of those with agreement tweets), and 5807 were messages that did not contain racist and/or xenophobic hate (56.46% of the total, 67.62% of those with agreement). A total of 16.5% of the sample was rejected for not having inter-judge agreement (n = 1697). Subsequently, the agreement of the manual classification with the predictions of the detector with the best performance was checked, and new evaluation metrics were extracted from the detector with that new sample to know

to what extent it had coincided with the human classifiers, and thus check how accurate and reliable had been the detection of the classifier with the new tweets.

*4.5. Evaluation of the Prototype on Multisource Content and in Other Languages*

An important objective of this research was to create a detector that could be capable of detecting hate speech in other similar contexts and languages. The ability of the models with the best performances to isolate hate speech coming from multiple online sources had to be investigated. Sources include Twitter, YouTube, Facebook, web articles, and comments from digital media, association websites, and blogs in which the type of hate speech analysed could be potentially spread. In addition to this, it is very important for the effectiveness of the proposed approach to be validated in other languages as well by simply retraining the model in the new target languages. For evaluating these desirable characteristics, we made use of the PHARM datasets [49]. The Preventing Hate Against Refugees and Migrants (PHARM) project concerns a multi-source platform for the analysis of unstructured news and social media messages. In addition to a web interface for scraping and analysing hate speech, PHARM offers a multilingual dataset containing multisource racist/xenophobic hate speech records. The sources include websites in Greek, Italian, and Spanish, as well as Twitter, YouTube, and Facebook, and concern news articles, comments, tweets, Facebook posts, and YouTube comments. Currently (September 2022), the PHARM dataset has about 35 k records. This dataset has been annotated by human coders, while the results have been checked for acceptable inter-coder reliability following the strategy previously developed in the Spanish prototype. In addition to the PHARM repository, the final datasets were augmented with additional data provided by the PHARM development team. Table 2 depicts the basic descriptive characteristics of the formed datasets.

**Table 2.** Dataset details for the supplementary evaluation in Spanish, Greek, and Italian.

|  | PHARM | | Other Sources | |
|---|---|---|---|---|
|  | **Hate** | **No Hate** | **Hate** | **No Hate** |
| ES | 1390 | 11,108 | 9727 | 23,787 |
| EL | 4359 | 6040 | - | 5362 |
| IT | 5848 | 4904 | - | 18,451 |

As will be further analysed in the following section, the most promising classifiers for the prototype hate speech detector proved to be the deep learning models. Therefore, this architecture was adopted for this experimental scenario as well, after making some slight modifications. The most notable modification was the addition of instance weighting in the training phase. Due to the imbalance between hateful and non-hateful records, the initial classifier tended to develop a bias towards the class that was overrepresented, the non-hateful class. The rest of the parameters remained the same. The number of units for each GRU layer was reduced to 64, as this setup offered the same performance with less computational cost.

## 5. Results

In the first place, before giving way to answering the research questions raised, we will explore the results of the manual annotation carried out to create the datasets. In this sense, the first thing to point out is the high percentage of tweets that were finally rejected, which is greater than 50%. This indicates the initial complexity to face the task of identifying this type of hate speech reliably and in a particular linguistic context. On the other hand, it is observed that the percentage of hate tweets with full agreement is considerably reduced despite having classified previously filtered messages. Specifically, and responding to RQ1, 15.6% of racist and/or xenophobic hate tweets (N = 3751) were validated, compared to 32.9% of messages labelled as non-racist/xenophobic hate (N = 7892). These percentages show that linguistic filter dictionaries, no matter how complete and complex, cannot be

an effective method to identify these types of hate messages, something that was already assumed. However, they served to optimize the process, since without these filters, the procedure of finding hate speech examples throughout Twitter would have been endless. In addition, considering that the datasets can always be enriched and updated with new data, the most important thing, especially in this initial prototype, was to establish a strategy to generate corpora of quality rather than quantity.

Regarding the evaluation of the generated models, three metrics were used: accuracy, F1-score, and AUC-ROC. The Accuracy was calculated by using the total sum of correct predictions across all classes; the F1-score was calculated as the arithmetic mean of the per-class F1-scores, which are the harmonic means of the precision and recall metrics. Finally, the AUC-ROC unveils the efficiency of the classifiers at all thresholds. Therefore, both micro- and macro-averaged metrics are present (accuracy and F1-score, respectively), with the latter being insensitive to a possible imbalance between classes. When macro-averaging, all classes are treated as equal, unveiling low scores on classes with few instances. As can be seen in Table 3, classification performance is at least acceptable in most cases, as scores higher than 0.75 have been recorded. Except for the simple Naïve Bayes algorithm, all models showed similar performances. Moreover, the accuracy and AUC-ROC scores were higher for the ad hoc RNN model, which confirms the comparative advantage of deep learning in this type of task. Responding to RQ2, we can affirm that logistic regression and support vector machines are the shallow algorithms that present the best performance in this case. Responding to RQ2A, we can confirm that the deep learning approach offers the highest performance. The results are also visualized in Figure 1.

**Table 3.** Evaluation metrics of the models generated with each of the algorithms.

| Classification Algorithm | Accuracy | F1-Score | AUC |
|---|---|---|---|
| NB | 0.67 | 0.73 | 0.65 |
| MNB | 0.75 | 0.83 | 0.64 |
| BNB | 0.68 | 0.81 | 0.50 |
| LR | 0.78 | 0.84 | 0.71 |
| SGD | 0.75 | 0.82 | 0.69 |
| SVC | 0.76 | 0.83 | 0.71 |
| MVE (Majority Voting Ensemble) | 0.76 | 0.82 | 0.68 |
| RNN | 0.86 | 0.78 | 0.92 |

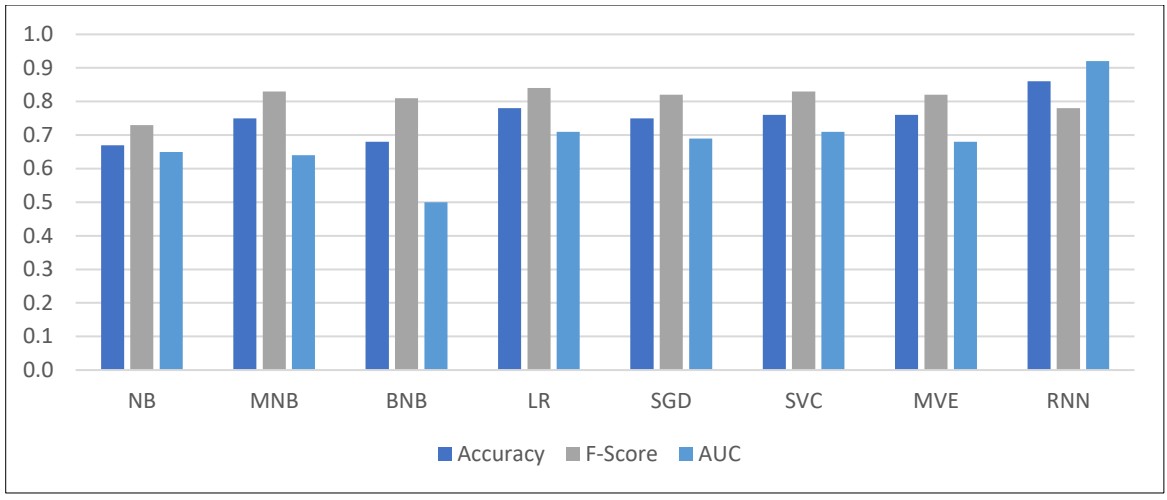

**Figure 1.** Performance ratings for all tested algorithms.

Next, the external validation phase was carried out. At this stage, the aim was to assess how capable the deep learning classifier generated ad hoc is on new data, i.e., new tweets concerning different temporal contexts. For this, only the tweets with an agreement resulting from the manual classification of the new sample were used (n = 8588). Thus, after running the model on that sample, firstly, Krippendor's Alpha was used to check the intercoder reliability between the manually coded tweets with agreement and the predictions offered by the prototype. The result of this reliability pretest was $\alpha$ = 0.6, an acceptable figure, but not too high. Subsequently, the evaluation metrics of the deep model were extracted when being run and tested on the new sample, with more promising results: Accuracy = 0.85, F1-Score = 0.74, and AUC-ROC = 0.88. Taking these metrics into account and responding to RQ3, we can confirm that the classification prototype shows an acceptable performance when tested with new, unseen, real data.

Finally, the evaluation of the deep learning architecture on new datasets and languages took place. As indicated, at this stage, not only the RNN model developed ad hoc was improved and tested, but also a new deep learning model based on BERT (bidirectional encoder representations from transformers), a machine learning technique based on transformers which is supposedly the most advanced linguistic model for natural language processing and especially for embedding contextualized words [48,50], was improved and tested.

Table 4 depicts the accuracy and F1-score metrics for the two deep models and the Spanish, Greek, and Italian languages. The results indicate that these models can have a good performance in different languages as well, not only in Spanish, and in multisource. Both models achieve high accuracy ratings consistently in the Spanish language (0.86, 0.85, 0.87, and 0.90 in the four tests, respectively) and show similar performances for the Greek and Italian languages, as well. It should be also noted that the gap between the micro- and macro-averaged metrics (accuracy and F1-score, respectively) became smaller, indicating that the instance weighting technique in the training process led to a better-balanced classifier without a bias towards the class with more paradigms. Therefore, responding to RQ4, we can confirm that the proposed deep learning architectures can be applied to detect racist and xenophobic hate speech in other languages and other platforms as well, retaining its performance.

**Table 4.** Evaluation metrics of the deep learning models trained on the PHARM datasets.

| Language | Accuracy | F1-Score |
|---|---|---|
| **AD-HOC RNN model** | | |
| ES | 0.87 | 0.87 |
| EL | 0.79 | 0.78 |
| IT | 0.91 | 0.89 |
| **BERT-based model** | | |
| ES | 0.90 | 0.86 |
| EL | 0.81 | 0.76 |
| IT | 0.91 | 0.88 |

## 6. Discussion

This work has generated the first prototypes capable of detecting anti-immigration online hate speech automatically and on a large scale. These classifiers were tested and validated firstly for messages spread on Twitter in Spanish, and later improved and adapted for their application on more online sources (YouTube, Facebook, and media websites) and in more languages (Greek and Italian). For this, we have generated ad hoc datasets through manual sorting tasks, and we have used, firstly, traditional classification algorithms for the generation of the primary models, and secondly, two different deep learning strategies, an innovation with respect to detectors, developed by other authors. Specifically, the

development of the different classifiers had been based on natural language processing and supervised machine learning techniques. Regarding the machine learning algorithms used, different shallow and deep architectures have been put to the test, from the most traditional based on shallow modelling, to deep models based on transformers, comparing their performances.

We have confirmed that deep modelling performs considerably better than shallow modelling for detecting hate speech directed towards migrants and refugees in tweets in Spanish since the models trained with neural networks were the ones that broadly presented the best evaluation metrics, something that had already been evidenced in numerous past studies [23,29,30]. That is why they were the ones used for the model's improvement and adaptation to detect anti-immigration online hate speech in new languages and sources, both offering acceptable performances in all new cases. At this point, it should also be noted that the transformer-based model seems to offer a slightly better performance in general terms than the ad hoc developed RNN model, something that was also expected since it is what the most recent studies showed [48,50]. Our findings comply with those of recent studies, such as in [51], where machine learning models (support vector machines and logistic regression), deep learning models (LSTM, CNN, and Bi-LSTM), and transformer-based language models (multilingual BERT, XLM, and monolingual Spanish BETO) are compared for hate speech detection in Spanish. The BERT-based models outperformed the ML and DL, with the BETO achieving the higher F1-score of 77.62%.

## 7. Conclusions

In way of conclusion, this research work proves that it is feasible to generate automatic detectors of online racist/xenophobic hate speech using machine learning with solid performances. In addition, specific validated datasets have been generated ad hoc for the training of the predictive models, something that other authors have not considered so far and that allows for the models to overcome the possible weaknesses derived from the internal validity of the detectors previously developed. In this sense, we can point out that, although the amount of hate and no hate messages added to the datasets after cross-coding and data cleaning may seem low, the most important thing in this process is to have quality examples over quantity. Moreover, this is because, although the evaluation metrics could be acceptable, if the examples are not completely reliable, the internal validity of the prototype could be contaminated with false positives or negatives. For this reason, we also wanted to go further and conduct an external validation to check the performance of our primary prototype more effectively when applied to new real examples. This validation with new data has served to verify that our deep model is reliable and has an acceptable performance when used in practice, with new real cases, and compared with a new validated manual classification. Nevertheless, from the beginning, the focus was on generating a quality and reliable dataset that can be used as a training corpus, since, in addition, the quantity can always be improved by including new validated examples into the corpus. In sum, we have resolved that logistic regression and support vector machines are the shallow algorithms that offered the best performance for this task out of the six tested. However, as indicated, we have confirmed that deep learning performs better than shallow learning for detecting racist and xenophobic online hate speech. The two tested architectures, the ad hoc developed RNN and the one based on transformers, presented considerably better performances than the traditional models, both in the Spanish language and on Twitter, as well as in other languages and sources. Nevertheless, it should be highlighted that the BERT-based model seems to offer a slightly better performance than the ad hoc RNN model, as expected.

In summary, it can be concluded that the results of this study are relevant, significant, and they represent a novelty in the scientific literature on the study of new computational methods applied to the social sciences and, specifically, to the detection and monitoring of online hate speech. This is so because, until now, anti-immigration online hate detectors had not been tested in Spanish, Greek, and Italian and in multisource using ad hoc developed

datasets and deep architectures, including an ad hoc RNN and a BERT-based model. So, it must be highlighted that this study offers a contribution in methodological terms, of course, with the large-scale detection strategy, with the generation of the ad hoc datasets of validated real examples of racist and xenophobic hate speech in Spanish, Greek, and Italian retrieved, firstly, from Twitter, but also and secondarily from YouTube, Facebook, and potentially hate-spreading websites, and lastly, with the models developed with both deep learning techniques. In addition, this research provides a theoretical advance in the study of online hate speech directed towards migrants and refugees. Finally, a practical and social contribution is also presented since the technology developed here can be applied in diverse public and private spheres, being able to benefit from private companies, research groups, nonprofit organizations, as well as government agencies, among other things, to draw the possible social acceptance or rejection of migrants in different European regions, and thus aid in executing strategies to improve long-term integration in migration processes. In fact, the prototypes that have been evaluated are already being used to develop new projects, such as the one recently published by Arcila-Calderón et al. [52] using the infrastructure and the computational strategy validated in the PHARM Project that allows for the retrieval of geolocated text messages from online sources using different search queries and criteria, and its direct and massive processing and classification through the Supercomputing Centre of Castilla y León, Scayle, where the models are executed to later generate datasets with the messages finally classified with reliability that can be consulted and analysed.

## 8. Limitations and Future Lines of Research

A major limitation of the approach comes from the fact that comments and tweets are treated as standalone texts without taking into consideration context information. Context information may rely on network analysis to determine the topic on which a comment or reply is poste and takes into consideration previous comments and replies that may be crucial to complete the meaning of the text that is analysed. However, the metadata that are collected along with the textual information may be useful for a more comprehensive analysis in the future. Furthermore, the detector should be usable for people who do not have computer skills or are not capable of running or manipulating a script without complications. With this purpose, a GUI integration has been presented in the work of Vrysis et al. [49]. This led to the creation of the PHARM web interface that incorporates the models that are presented in the current paper, providing a visual and friendly interface that allows for the use of the detectors by nonexperts, universalizing its use and applications. This can also broaden the range of practical possibilities as well as social benefits since the detector could be implemented by more social actors. There is strong evidence that the robustness and generalization capabilities of deep learning models depend highly on the quality and quantity of available data. The widespread use of the PHARM interface is expected to lead to an extension in the dataset, since it may allow the retraining of models in the future following the presented methodology. Finally, the future possibility of implementing an early-warning system is raised. This could warn about increases in racist/xenophobic hate speech in a geo-localized way and thus prevent and predict possible increases in racist/xenophobic hate crimes in certain regions. This warning system could also take advantage of the interface mentioned above to be visual and, in the same way, usable by all types of users.

**Author Contributions:** Conceptualization, C.A.-C. and J.J.A.; methodology, C.A.-C., J.J.A., L.V. and N.V.; validation, J.J.A., L.V. and N.V.; formal analysis, J.J.A., L.V. and N.V.; investigation, J.J.A., P.S.-H., M.O.A., L.V. and N.V.; resources, C.A.-C.; writing—original draft preparation, C.A.-C. and J.J.A.; project administration, P.S.-H.; funding acquisition, C.A.-C. All authors have read and agreed to the published version of the manuscript.

**Funding:** This research was funded by the European Union through the Rights, Equality, and Citizenship programme REC-RRAC-RACI-AG-2019 [GA n. 875217].

**Institutional Review Board Statement:** Not applicable. In this study, authors have only worked with public data extracted from the Twitter API v2. Therefore, the authors consider that there is no ethical conflict that needs approval.

**Informed Consent Statement:** Not applicable. No person external to the project participated in this study, so no consent was necessary.

**Data Availability Statement:** All the material used in this work will be available on the project website: https://pharmproject.usal.es/, accessed on 8 September 2022.

**Acknowledgments:** Andreas Veglis and Sergio Splendore.

**Conflicts of Interest:** The authors declare no conflict of interest in this work.

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
