# Peer review of "How to Detect Online Hate towards Migrants and Refugees? Developing and Evaluating a Classifier of Racist and Xenophobic Hate Speech Using Shallow and Deep Learning"

_sustainability, doi:10.3390/su142013094_

Round 1

Reviewer 1 Report

The only concern I have about the study was mentioned as a limitation of the study by the authors (the importance of context in identifying hate speech etc). I also think that the authors have addressed my concern and for this reason, I have no further issue with the study. I think that it is truly amazing for technology to identify hate speech (racist, xenophobia comments) for the establishment of an early warning system. Accuracy of such detection needs further improvement as inaccurate identification of hate speech will be seen as an effort to curb freedom of speech. Nevertheless the results reported in the study are highly encouraging.

Author Response

We thank reviewer 1 for his/her positive review.

Reviewer 2 Report

This paper needs extensive revision, as follows:

-Almost, no model description was presented in this paper. So, add detailed descriptions about your model supported by math definitions, as well as pseudocodes.

- Describe well the settings of all compared models, and try to include new methods from recent years in your comparisons.

- More evaluation metrics and tests are needed.

- The complexity also needed to assess the quality of the model.

Refresh your paper with most recent studies, such as:

Arabic Offensive and Hate Speech Detection Using a Cross-Corpora Multi-Task Learning Model;

Social network conversations with young authors of online hate speech against migrants;

Evaluating transfer learning approach for detecting Arabic anti-refugee/migrant speech on social media;

Author Response

The authors appreciate referee's comments and suggestions for improvement. Below is a table in which the authors explain point by point the changes and improvements made to the manuscript, and respond to the referees' comments. In the case of not having been able to address the suggested changes, the reasons are also explained in the responses to the referees.

REVIEWER 2

AUTHORS

This paper needs extensive revision, as follows:

- Almost, no model description was presented in this paper. So, add detailed descriptions about your model supported by math definitions, as well as pseudocodes.

Some more descriptions about the deployed deep-learning models were added in Section 4.3.2. Additional information about the shallow-learning models was added in Section 4.3.1 as well. The models employed in the study are well-known/established algorithms that have been presented to and used by the scientific community. More information about the RNN and BERT models has been documented in Cho et al. (2014) and Devlin et al. (2018). The references are included in the revised manuscript. Concerning the baseline models, the default configuration provided by the sci-kit library was selected, we added this information to the manuscript along with a reference to the official documentation.

- Describe well the settings of all compared models, and try to include new methods from recent years in your comparisons.

State-of-the-art deep-learning approaches were deployed as part of the experimental setup. In specific, an RNN (with GRUs) and a BERT model were evaluated. Our findings agree with those of recent studies and the following text has been added to the manuscript, as well as additional references (Plaza-del-Arco et al., 2021). The following text was added to the manuscript:

Our findings comply with those of recent studies, such as in Plaza-del-Arco et al. (2021), where machine learning models (Support Vector Machine and Logistic Regression), deep learning models (LSTM, CNN and Bi-LSTM ) and transformer-based language models (multilingual BERT, XLM and monolingual Spanish BETO) are compared for hate speech detection in Spanish. The BERT-based models outperformed the ML and DL, with the BETO achieving the higher F1-Score of 77.62%.

- More evaluation metrics and tests are needed.

A complete set of metrics (Accuracy, F1-Score, and AUC) has been reported and documented in the manuscript. Nine different models were examined in total and the most capable ones were tested in three different languages. In addition to this, as aforementioned in the previous answer, the results are in line with recent studies (Plaza-del-Arco et al., 2021; Aldjanabi et al., 2021; Mohdeb et al., 2022).

- The complexity is also needed to assess the quality of the model.

We have provided more details concerning the complexity of the evaluated models in the revised manuscript:

Table 1. Deep learning algorithms complexity in terms of models’ number of parameters

RNN

BERT

216,798

177,854,978

Refresh your paper with most recent studies, such as:

●      Arabic Offensive and Hate Speech Detection Using a Cross-Corpora Multi-Task Learning Model;

●      Social network conversations with young authors of online hate speech against migrants;

●      Evaluating transfer learning approach for detecting Arabic anti-refugee/migrant speech on social media;

We would like to thank the reviewer for their suggestions. All the studies have been included in the revised manuscript:

Studies indicate the phenomenon of de-responsibilisation (Pasta et al., 2022) of hate speech spreading online, especially by younger audiences who feel that their public hate language should not be taken seriously.

[...] In Mohdeb et al. (2022) transfer learning is investigated, introducing the BERT-based transformer AraBERT that shows improved performance in Algerian dialectal Arabic. In [NR3], multi-task learning (MTL) is proposed, for the adaptation of a pre-trained hate speech detection model in the Arabic language in cross-corpora tasks.

Reviewer 3 Report

The topic is interesting and very relevant to the current context, moreover, the paper has wider theoretical and practical applications. The authors have put their best efforts to execute this paper. However, I have the following reservations and suggestions for the sake of improvement of the undertaken study:

1) The logical sequence of the abstract should be as 1) objectives, 2) methodology, 3) Findings, 4) conclusion and 5) implications. Thus, the authors should also rewrite the abstract in this sequence. The authors should mention the analysis techniques in the methodology, After the findings conclusion, and then please describe important implications. 

2) The authors did not establish the motivation, significance, and novelty of the undertaken study. The authors are suggested to improve this important factor in the "Introduction" section. The background of the research should also be presented in the section.

3) The literature should be presented separately from the introduction, and it should be presented in an audit form and should be linked with the objectives of the current paper.

4) The discussions section provides the opportunity for the authors to sell their idea to the readers. The discussions are separated from the conclusions section, and it should be complemented with the previous literature.

5) The conclusion should be added after the discussions section, the conclusion is always one step ahead of the findings. 

6) The practical, theoretical and societal implications should be discussed after the conclusion, and in the light of the conclusion and discussions. 

Author Response

The authors appreciate referee's comments and suggestions for improvement. Below is a table in which the authors explain point by point the changes and improvements made to the manuscript, and respond to the referees' comments. In the case of not having been able to address the suggested changes, the reasons are also explained in the responses to the referees.

REVIEWER 3

AUTHORS

1) The logical sequence of the abstract should be as 1) objectives, 2) methodology, 3) Findings, 4) conclusion and 5) implications. Thus, the authors should also rewrite the abstract in this sequence. The authors should mention the analysis techniques in the methodology, After the findings conclusion, and then please describe important implications.

We appreciate the suggestions for improvement by the reviewer. We have rewritten the abstract following the structure indicated:

[...] The objective of this research is to overcome the limitations of those previous works, developing and evaluating classification models capable of detecting racist and/or xenophobic hate speech spread online, first in Spanish, and later in Greek and Italian. In the development of these prototypes, three differentiated machine learning strategies are tested. First, various traditional shallow learning algorithms are used. Second, deep learning is used, in specific an ad-hoc developed RNN model. And finally, a BERT-based model is developed, in which Transformers and neural networks are used. The results confirm that deep learning strategies perform better in detecting anti-immigration hate speech online. Of both deep architectures, moreover, it is the one developed with BERT that shows better evaluation metrics. It is for this reason that the deep models were the ones finally improved and tested for hate speech detection in Greek and Italian and in multisource. The results of this study represent an advance in the scientific literature in this field of research, since up to now no online anti-immigration hate detectors had been tested in these languages, and using this type of deep architectures.

2) The authors did not establish the motivation, significance, and novelty of the undertaken study. The authors are suggested to improve this important factor in the "Introduction" section. The background of the research should also be presented in the section.

Following the instructions of the reviewer, we have included a paragraph explaining the significance and relevance of the study, as well as the novelties compared to previous works. This has been included in the third paragraph of the introduction:

For these reasons, the objective of this work is to develop and evaluate a detector of racist and xenophobic hate speech online. The novelties of this work compared to the previous ones are several. In the first place, for the development of the prototypes we will generate ad-hoc databases with messages extracted from social media and manually annotated by several trained coders. Second, in addition to shallow learning algorithms, deep architectures will also be used in the development of the prototypes. On the one hand, an ad-hoc developed RNN model, and, on the other hand, a BERT-based model, in which Transformers and neural networks will be used. Finally, although the prototypes will be trained and tested first in the Spanish language, the ones with the best performance will later be improved and retrained following the same strategy so that they are capable of detecting anti-immigration hate speech also in Greek and Italian (Mediterranean countries that are the main gateway for most immigrants arriving in Europe) and in multiple sources.

Regarding the background, we consider that this has already been explained, together with the context and justification of the work, in the first two paragraphs of the introduction.

3) The literature should be presented separately from the introduction, and it should be presented in an audit form and should be linked with the objectives of the current paper.

We agree with the reviewer and welcome their comment, but we think that this has indeed already been considered when writing the previous version of the manuscript. We think that the literature review and state of the art presented is closely linked to the objectives of the paper, and it has already been presented correctly and separately from the introduction, in sections 2 and 3:

2. Defining Online Hate Speech

3. Detecting racist and xenophobic hate speech online

4) The discussions section provides the opportunity for the authors to sell their idea to the readers. The discussions are separated from the conclusions section, and it should be complemented with the previous literature.

Addressing the reviewer's indications, we have separated the discussion from the conclusions, and we have expanded this discussion, linking the results of this work with the previous related literature. It can be consulted in the new section 6.

5) The conclusion should be added after the discussions section, the conclusion is always one step ahead of the findings.

In the same way, the conclusions have been added later to the discussion, in a new section below, which has been numbered as 7.

6) The practical, theoretical and societal implications should be discussed after the conclusion, and in the light of the conclusion and discussions.

Following the suggestions, at the end of this new section of conclusions we have included the practical, theoretical and societal implications in the light of the conclusion and discussions.

Round 2

Reviewer 2 Report

The authors addressed all comments raised in the previous rounds. This version is ready for publication. So, it can be accepted in its present form.